# Platelet-Leukocyte Interplay in Cancer Development and Progression

**DOI:** 10.3390/cells9040855

**Published:** 2020-04-01

**Authors:** Dagmar Stoiber, Alice Assinger

**Affiliations:** 1Department of Physiology and Pharmacology, Medical University of Vienna, 1090 Vienna, Austria; dagmar.stoiber@kl.ac.at; 2Department Pharmacology, Physiology and Microbiology, Division Pharmacology, Karl Landsteiner University of Health Sciences, 3500 Krems, Austria

**Keywords:** platelets, cancer, platelet-leukocyte aggregates, inflammation

## Abstract

Beyond their crucial role in hemostasis, platelets are increasingly recognized as regulators of inflammation. Via modulation of the immune system by direct and indirect interactions with leukocytes, platelets regulate several aspects of tumor-associated pathology. They influence inflammatory processes in cancer at various stages: platelets alter the activation status of the endothelium, recruit leukocytes to tumor sites and attune the inflammatory milieu at sites of primary and metastatic tumors. Patients with cancer show systemic changes of platelet activation. Tumor-associated platelet activation facilitates initiation of the coagulation cascade and constitutes a significant risk for thrombosis. Tumor-activated platelets further contribute to cancer progression by promoting critical processes such as angiogenesis and metastasis. Platelets modulate innate leukocyte effector functions such as antigen presentation by dendritic cells, monocyte recruitment and differentiation or neutrophil extracellular trap formation, which sculpture immune responses but also promote thrombosis and metastasis. On the other hand, responses of the adaptive immune system are also regulated by platelets. They are also involved in T-helper cell 17 differentiation, which represents a double-edged sword in cancer progression, as these cells propagate angiogenesis and immunosuppressive activities but are also involved in recruiting immune cells into tumors and stimulating effector CD8^+^ T cells. Moreover, platelets fine-tune tumor surveillance processes by modulating natural killer cell-mediated cancer cell recognition and effector functions. This review aims at summarizing the role of platelet-leukocyte interactions in the development and progression of cancer and puts its focus on cancer-related alterations of platelet and leukocyte functions and their impact on cancer pathology.

## 1. Platelet-Leukocyte Interactions

Platelets are anucleated cell fragments of megakaryocytes that prevent excessive blood loss upon vessel injury by a process termed hemostasis. Under physiologic conditions, the resting endothelium constantly releases nitric oxide (NO) and prostacyclin (PGI_2_) into the circulation to keep platelets in a resting state [1]. Upon endothelial damage, proteins of the subendothelial matrix are exposed into the lumen, resulting in binding and activation of platelets [1], which then recruit further platelets, aggregate and seal the wound.

However, platelets also become activated during inflammatory conditions, leading to platelet interactions with endothelial cells and leukocytes. More and more, evidence emerges that platelets thereby also fulfil essential roles in immunity and modulate physiologic and pathologic responses during inflammation and infection. Upon activation, platelets rapidly interact with innate immune cells and exert immunomodulatory effects directly via cell-cell contact and/or indirectly via the release of chemokines and cytokines [2]. Platelet-leukocyte interactions result in leukocyte recruitment, activation, extravasation, phenotype switch and changes in effector functions [3]. Crosstalk between platelets and leukocytes is initiated by P-selectin (CD62P), which gets expressed on the platelet surface upon activation and interaction with P-selectin-binding glycoprotein 1 (PSGL-1) on the surface of leukocytes. This interaction is further stabilized by platelet GPIb, GPIIb/IIIa, ICAM-2 and CD40L, which either interact directly with respective receptors on leukocytes or via bridging by fibrinogen or extracellular matrix components (e.g. collagen or laminin). This direct platelet-leukocyte interaction fosters mutual activation and triggers the release of platelet granule contents (including platelet factor 4 (PF4/CXCL4), stromal cell-derived factor 1 (SDF-1), CD40L and CD62P) and generation of lipid mediators (e.g. thromboxane A2 (TxA2)), which modulates various leukocyte effector functions [3,4,5]. Due to differences in the affinity of platelet P-selectin and PSGL-1 on different types of leukocytes [6], platelets preferentially bind monocytes over neutrophils [7]. These heterotypic complexes are detected in the blood of patients suffering from various thrombotic or inflammatory conditions [8] and lead to modulations of leukocyte phenotype and functions but, also, affect tissue repair processes and angiogenesis. Thereby, platelets and platelets-leukocyte interactions are involved in various different diseases, including several steps of cancer development and progression.

## 2. Platelet-Leukocyte Interactions in Tumor Development

The development of cancer is a multistep process that depends on several hallmarks such as sustaining proliferative signaling, evading growth suppressors, resisting cell death, enabling replicative immortality, inducing angiogenesis, activating invasion and metastasis, reprogramming of energy metabolism and evading immune destruction [9,10]. A prerequisite for the occurrence of these processes is genome instability, which generates genetic diversity. These somatic mutations represent specific signatures that lie at the root of the distinct cancer types [11] and provide proof for the concept of cancer as a complex and heterogeneous disease. Inflammation fosters several of these hallmark functions [9]. Thus, tumor development depends on the complex interplay of mutagenized tumor cells with their local and distant microenvironment [12,13], and herein, platelet-leukocyte interactions play an important role.

### 2.1. Inflammation and Cancer

It is a well-accepted notion that chronic inflammation and cancer development are intertwined processes, and about one-fifth of all human cancers are caused by infections, exposure to irritants or autoimmune disease [14]. An inflammatory tumor immune micromilieu is a premise of all tumors, even in those that do not develop during chronic inflammation. Interestingly, organs with high tumor incidence in the context of chronic inflammation are those that usually have close contact with microbial products or directly with microorganisms, further pointing to the crucial role of the inflammatory tumor microenvironment [15].

Pathogens are commonly recognized via conserved molecular structures by toll-like receptors (TLRs) on immune cells, as well as other cells (including platelets). TLR-binding triggers the expression of several adapter proteins and downstream kinases, inducing expression of key pro-inflammatory mediators, activation of the innate immune response and maturation of the adaptive immune components. During cancerogenesis, the role of TLRs and their ligands becomes more complex as they exert context-specific roles by either suppressing tumor growth or promoting survival of malignant cells. Several TLR agonists are currently tested for their effects on tumor therapy. Agonists for, e.g. TLR3 have been shown to induce apoptosis of tumor cells [16], and recently, a TLR2 agonist that generates macrophages with strong anti-tumor potential has been discovered [17].

Inflammatory cells and their mediators (including cytokines, chemokines and prostaglandins) in the tumor microenvironment are involved in various proinflammatory responses, which can act in an autocrine and/or paracrine manner on malignant and nonmalignant cells [18]. The composition of immune cells and other components of the tumor microenvironment are crucial for the progression and, also, the outcome of this malignancy. Essentially, all immune cells can play a role in the microenvironment during tumor development and progression.

Platelets become activated upon injury, direct contact with pathogens via their TLRs or indirectly by inflammatory conditions, which results in the release of growth factors and the modulation of immune response [3,19], thereby further contributing to tumor-sculpting processes. Additionally, tumor cells can activate platelets through secretion of factors such as ADP [20], which, in turn, leads to degranulation and release of proangiogenic and protumorigenic factors [21] (Figure 1). Platelets bind tumor cells via C-type lectin-like immune receptor 2 (CLEC-2) interaction with podoplanin, which is expressed at the invasive front of many tumors and associated with poor outcome in various cancer types. Via CLEC-2, platelets promote tumor progression, metastasis and cancer-induced thrombosis. Podoplanin-CLEC-2 interaction further enhances the immunosuppressed microenvironment, facilitating spread and growth of the cancer [22]. Further, high-mobility group box 1 (HMGB1), released by dying tumor cells, interacts with TLR4 on platelets and mediates platelet–tumor cell interaction, which promotes metastasis [23]. In addition, the extracellular matrix gets exposed during tumor cell invasion or as circulating cancer cells interact directly with platelets, leading to platelet activation and release of proadhesive and proangiogenic factors that facilitate cancer progression [24].

In addition, platelets cooperate with leukocytes in the tumor microenvironment (Figure 1) as various types of immune cells are recruited to the site of action during tumor development. The recruitment occurs in response to continuous crosstalk of tumor and immune cells, and this also explains the contrasting functions—host-protective and tumor sculpting—of the immune system in cancer [25]. Myeloid cells play an essential role in the regulation of immune functions. They eliminate aberrant cells and present the acquired antigens to T cells. Depending on the myeloid cell phenotype, this interaction induces either effector cytotoxic lymphocytes or regulatory T cells and, thus, determines the immune response against the tumor. Myeloid cells also regulate tissue growth, homeostasis, repair and remodeling via their expression of numerous cytokines, chemokines, growth factors, proteolytic enzymes and scavenger receptors. Platelets have been demonstrated to orchestrate myeloid cell functions by enhancing leukocyte extravasation and modulation of cytokines released [3]. Although direct platelet-leukocyte interactions have never been demonstrated at the onset of tumorigenesis, we know from various in vivo and in vitro studies that platelets get activated by tumor cells and that their immune-regulatory functions are very likely to occur already at the onset of cancer development.

Effector cells with the capacity to destroy transformed cells in early tumor development are natural killer (NK) cells and cytotoxic T lymphocytes (CTLs). Eventually, some tumor cells overcome immune destruction and undergo immunoediting, i.e., sculpting by the immune system. Sculpted tumors then progress, become clinically apparent and induce an immunosuppressive tumor microenvironment [25]. There, regulatory T (Treg) cells and myeloid-derived suppressor cells (MDSC) dominate the scene and enable tumor escape and progression. The role of platelet-leukocyte interactions in this context is currently unclear. However, there are hints from other diseases that platelets can modulate effector functions in a proinflammatory environment, and some factors that are crucial in cancerogenesis are also modulated by platelets in other diseases.

Among the noncellular mediators involved in tumor development, interleukin-1 (IL-1) plays a central role in inflammation. IL-1 is a member of a family of structurally related cytokines and exists as two isoforms of IL-1 (α and β). IL-1α and IL-1β are responsible for a broad spectrum of immune and inflammatory responses and essential drivers for the activation of NFκB–regulated genes, including cytokines and chemokines required for the establishment of a protumoral microenvironment [26]. IL-1β further induces the formation of platelet-leukocyte aggregates and augments both megakaryocyte and platelet functions, thereby promoting a prothrombotic environment and potentially contributing to thrombotic diseases related to cancer [27]. Platelets further modulate monocyte functions via direct interaction and promote a CD16-positive phenotype [28,29], which is associated with inflammatory features, including the release of IL-1β [30]. Therefore, several agents that block or neutralize the IL-1 signaling pathway are utilized or being tested for tumor treatment [15].

Another important inflammatory cytokine is IL-17, which is produced by a subtype of T helper cells known as T helper 17 (Th17) cells and which acts as a double-edged sword in cancer [31]. Initially, IL-17 was considered to promote neovascularization and tumor cell proliferation, as well as immunosuppressive activities. However, IL-17 also activates cytotoxic T cells, NK cells and neutrophils; fosters a Th1 phenotype and promotes interferon (IFN)-γ production [32,33,34]. Platelets play an important role in T cell homeostasis as platelet CXCL4 limits Th17 cell differentiation [35], and platelet-derived microvesicles were shown to inhibit IL-17 production by Treg cells through P-selectin [36]. Thereby, platelets modulate immunosurveillance processes during cancer development.

A key multifunctional cytokine with pleiotropic actions in the regulation of immune responses associated with inflammation is TNF-α. Biological agents targeting its signaling cascade are commonly used to treat inflammatory conditions or autoimmune diseases. Both tumor and stromal cells of solid tumors secrete TNF-α and, interestingly, tumor promoting but also inhibitory effects were described [14]. However, the role of platelets on TNF-α production is controversial. While, during infection, platelets diminish TNF-α release by macrophages, platelets also render monocytes towards a CD16-positive phenotype, which is associated with enhanced TNF-α production [37].

Important chemokines frequently targeted in inflammation and cancer are CCL2, IL-8 and CXCL12. CCL2 is highly expressed in tumor but also stromal cells of many cancers. It is speculated that CCL2 directly stimulates tumor cell proliferation, survival and migration; influences angiogenesis and acts as a chemotactic factor to tumor cells and inflammatory monocytes. CCL2 is mainly produced by endothelial and smooth muscle cells, and activated platelets have been demonstrated to enhance CCL2 expression on both cell types [38].

IL-8 (CXCL8) is known as a chemotactic factor for T cells, neutrophils and basophils and has paradoxically both tumor-promoting and -inhibitory functions. Adhesion of monocytes to platelets—a common event under inflammatory conditions—results in the translocation of NFκB to the nucleus, where it triggers IL-8 expression by monocytes [37,39]. CXCL12 signals via the cell surface receptor CXCR4 and was identified in lymphocytes trafficking to the bone marrow. CXCR4 is highly expressed on malignant cells in many different tumor types [14]. Platelets in turn get activated by CXCL12 but also produce CXCL12, which has important implications for monocyte differentiation and neointima formation [38,40].

Recently, antibody-based immunotherapies targeting the so-called immune checkpoints, such as PD1/PDL1, were clinically approved in anticancer therapy. Immune checkpoints downmodulate immune functions. PD1 represents a surface molecule on T cells (but also NK cells). It plays a role in their exhaustion, which is driven by inflammatory cytokines such as interferon-γ (IFN-γ) secreted by tumor antigen-specific T cells [41]. Antibodies targeting immune checkpoints demonstrate that the immune system can be harnessed for tumor treatment, as the response to anticancer therapy is largely influenced by the crosstalk of tumor cells with their microenvironment [42].

PDL1, the ligand for PD1, is also expressed on human platelets and is affected by immune checkpoint therapy [43]. Immune checkpoint inhibitors thereby bear immune-related hematological toxicity, such as immune-related thrombocytopenia [44]. Recently, it was shown that platelets conjugated with PDL1, as well as genetically engineered PD1-positive platelets, accumulate within the tumor surgical wound and revert exhausted CD8^+^ T cells. This led to the eradication of residual tumor cells in a mouse model. Further, platelets loaded with low-dose cyclophosphamide were able to deplete regulatory T cells, thereby preventing tumor relapse [45]. These recent data indicate that platelets can also be targeted to optimize tumor therapy. Indeed, platelets have previously been shown to represent optimal tools for the targeted and controlled release of therapeutics superior to nanoparticle systems, and platelet-based drug delivery was shown to improve drug tissue enrichment while reducing adverse effects in murine models [46,47].

### 2.2. Platelet-Leukocyte Interactions in Virus-Associated Cancers

Infectious cancer agents can be divided into two broad categories: direct carcinogens, which express, e.g. viral oncogenes that directly contribute to cell transformation, and indirect carcinogens, which cause cancer through chronic inflammation, eventually resulting in carcinogenic mutations. It is estimated that 15% of all cancers are caused by viral infections. In humans, seven viruses are known to promote cancer: (1) Epstein-Barr virus (EBV), which is linked to Burkitt’s lymphoma and nasopharyngeal carcinoma, lymphoproliferative disorders, Hodgkin’s disease, non-Hodgkin’s lymphoma and some gastrointestinal lymphoma; (2 and 3) hepatitis B and C virus (HBV and HCV), which lead to hepatocellular carcinoma; (4) human T-lymphotropic virus-I (HTLV-I), which causes adult T cell leukemia; (5) human papillomaviruses (HPV) 16 and 18, causing cervical, penile and some other anogenital cancers, as well as head and neck cancer; (6) Kaposi’s sarcoma herpesvirus (KSHV), causing Kaposi’s sarcoma, primary effusion lymphoma and some multicentric Castleman’s disease amd (7) Merkel cell polyomavirus (MCV), resulting in Merkel cell carcinoma [48].

Generally, platelets act as a double-edged sword in viral infections as they directly and indirectly suppress infection but, also, to support virus persistence in response to certain viruses [49]. Platelets influence viral dissemination by protecting viruses from recognition by immune cells and foster viral attachment. While no direct interaction of platelets and some of these oncoviruses is known, HTLV infection leads to an increase in platelet as well as lymphocyte count, suggesting viral effects on hematopoietic growth factors or cytokines [50]. Further, platelets interact with EBV via complement receptor 2 (CR2), which results in the release of TGF-β [49].

Platelets also interact with HBV, which is responsible for more than 50% of hepatocellular carcinoma (HCC) cases worldwide, making it the second-most important known carcinogen for all types of cancer. HCC begins in the main type of liver cell (hepatocyte) and represents the most common type of liver cancer and, also, the fastest-raising cancer entity. HCC is a highly lethal tumor and, currently, the third-leading cause of cancer-related deaths [51]. During HBV infection, platelets adhere to sinusoidal hyaluronan via CD44 and scavenge CD8^+^ T cells, which then crawl along the liver sinusoids and scan hepatocytes for the presence of antigens. Hepatocellular antigen recognition triggers effector functions, which enhances viral clearance but, also, tissue damage [52]. These immune surveillance processes might get disturbed by liver fibrosis [53]. Antiplatelet drugs (aspirin and clopidogrel) potently reduce HCC development as they lead to a decreased number of HBV-specific CD8^+^ and nonspecific inflammatory cells that accumulate in the liver. However, since these cells are essential to control viral infections, antiplatelet therapy potentially accelerates HBV replication. This clearly shows that antiplatelet therapy should be well-monitored and combined with antiviral treatment [54]. Of note, in a large-scale, population-based cohort study, aspirin reduced the risk of HCC in patients affected by chronic liver disease of unspecified etiologies [55]. However, further trials are warranted to confirm this observation and unravel the underlying mechanism.

### 2.3. Platelet-Leukocyte Interactions and Thrombosis in Cancer

There is a strong association between cancer and thrombotic complications. Approximately 20–30% of all first venous thromboembolic events can be attributed to cancer [56]. Tumor patients have a four- to seven-fold increased risk for venous thromboembolism. However, the average risk of thrombosis, which is estimated to be 13 per 1000 person/years, strongly varies between cancer types. Therefore, cancers can be classified as high-risk (pancreas or brain, where incidents are as high as 200 per 1000 person/years), moderate (lung or colon) and low (prostate or breast). Further, in metastatic diseases and in patients receiving high-risk treatment (surgery or chemotherapy), there is an elevated risk of thrombotic complications [57]. Cancer-associated thrombosis is linked to high morbidity and mortality and is the second cause of death after cancer itself [58]. While the association between cancer and thrombosis is very clear, the mechanisms underlying the enhanced thrombosis risk remain unclear, and several different mechanisms have been proposed. The association between cancer and venous thrombosis is stronger than between cancer and arterial thrombosis, indicating that changes of the coagulation system rather than disturbed platelet functions are mainly responsible for the observed effects.

In vivo coagulation classically gets initiated by interaction of the tissue factor (TF) and coagulation factors. TF is one of the driving forces of microthrombus formation, proven by the fact that mice with pharmacologically or genetically impaired TF were protected from disseminated intravascular coagulation (DIC) in an endotoxemia model [59]. However, mice with impaired TF activity showed an increased bacterial burden during acute infection [60,61], indicating a crucial role of coagulation in immune defense processes. Many tumor cells express TF or other procoagulant substances [62], and also, the endothelium lining in the tumors shows upregulation in TF expression and increased vascular permeability [63]. Normally, TF is expressed in an encrypted form on the cell membrane. Platelets play a pivotal role for the activation status of TF (Figure 2). Direct interaction with platelets and leukocytes expressing TF enhances TF activity. Interactions and fusion events that typically occur between platelets and monocytes enhance the generation of monocyte- and platelet-derived microvesicles and their hybrids, fostering decrypted TF activity [64], which also contributes to thrombus formation [65,66]. Cancer patients show elevated levels of circulating TF-positive microvesicles of hematopoietic origin, as well as cancer cell-derived microvesicles [67,68].

Thrombus formation in larger vessels, like atherothrombosis and deep-vein thrombosis (DVT), may also be propagated by immunothrombotic events [69,70]. Especially, inflammation induced TF expression on innate immune cells is a major contributor to DVT [71]. Moreover, leukocytes and their associated microparticles are responsible for degradation of the TF pathway inhibitor (TFPI), and thereby, they accelerate TF-induced thrombotic events even more [72]. Further, platelets get activated by tumor cells themselves or the proinflammatory tumor microenvironment. During aggregation, platelets release a plethora of chemokines that support the recruitment of leukocytes to the site of injury [73]. They also secrete the protein disulphide isomerase, which supports the activation of leukocyte-derived TF [74].

Besides TF, another driving force in venous thromboembolism represents neutrophil extracellular trap (NET) formation. Interaction of platelet P-selectin with neutrophil P-selectin glycoprotein ligand 1 (PSGL-1) leads to neutrophil activation and mediates migration and NET formation [75]. During NET formation, neutrophils expel their DNA, which forms an insoluble net that captures and kills bacteria with antimicrobial proteins and provides a platform where leukocytes can act more efficiently [76]. NETs, on the other hand, recruit platelets by presenting histones to platelet TLR4, leading to their activation and causing again a procoagulant milieu that promotes thrombosis [77,78]. Although never directly shown in cancer patients, NET formation has been proposed as an underlying mechanism of cancer-mediated thrombosis. Tumor cells or tumor cell-derived microparticles trigger NET formation either directly or indirectly by priming platelets, which results in further platelet activation and TF release [79]. Most recently, clinical evidence is corroborating the association between NET formation and thrombosis in cancer patients. These data suggest that biomarkers of NET formation are associated with the occurrence of venous thromboembolism (VTE) in cancer patients, indicating a role of NETs in the pathogenesis of cancer-associated thrombosis [80]. Of note, NETs have also been shown to be responsible for awakening dormant cancer, thereby fostering a procancerogenic microenvironment [81].

Apart from immunothrombotic events, the tumor microenvironment leads to exposure of extracellular matrix proteins, which triggers platelet activation and fosters the recruitment of leukocytes to the sites of tumors. Moreover, tumor cells themselves express molecules such as CLEC-2 and integrins (e.g. RGD-binding integrins like αvβ3, αvβ5, αvβ6, αvβ8, α5β1 and α8β1) that foster platelet activation and, therefore, generate a prothrombotic milieu. Thus, expression of these receptors correlates well with metastasis and poor patient prognosis [82]. These integrins in turn allow binding to extracellular matrix proteins and fibrinogen, which enhances their adhesiveness and, thus, promotes angiogenesis. During metastasis, it is possible to observe a fibrin(ogen) mesh work around tumor cells, which also shields the tumor from attack by the inflammatory system [34]. Moreover, microthrombi consisting of fibrin-encapsulated metastatic cells can lodge in different tissues, providing a possible mechanism for the transport of metastatic cells through the vasculature [68].

## 3. Platelet-Leukocyte Interactions in Tumor Dissemination

### 3.1. Tumor Dissemination

Almost all metastasizing tumor cells reach distant sites in the body via the bloodstream. In the circulatory system, several stressors affect tumor cells and influence their survival and spreading: (i) loss of the integrin-mediated adhesion to the extracellular matrix, which can lead to programmed cell death (also known as anoikis), (ii) hemodynamic shear forces within the bloodstream and distortion of tumor cells due to capillaries of smaller diameters and, most importantly, (iii) elimination by the immune system. As solid tumors frequently present with an immunosuppressive microenvironment, tumor cells that depart from the primary tumor are suddenly confronted with—and often also eliminated by—a functional immune system. Thus, only a small amount of tumor cells successfully extravasates and forms metastasis [83].

Numerous metastatic tumors illustrate that at least a minority of cancer cells are able to withstand these stress factors. Some features that help the tumor cells to protect themselves against the different stressors in the bloodstream are cell autonomous (e.g. the overexpression of a truncated version of pannexin-1, a transmembrane channel protein, in combination with its wild-type counterpart, increases ATP release from the channel upon mechanical stress, which, in turn, leads to an autocrine loop inhibiting tumor cell apoptosis, as shown for breast cancer metastasis models [84]). However, most mechanisms require interactions with other cells, such as platelets and myeloid cells.

Tumor cells usually are larger in diameter than small capillaries and, therefore, are likely to become trapped therein [85]. This often leads to the clustering of tumor cells, and even those clusters can pass capillaries by the rapid formation of cell chains and transient deformation of single cells [86]. Mechanisms directed against the mechanical stress-induced death of tumor cells also include the formation of microaggregates with platelets in the microvasculature, which also protects the tumor cells in the blood stream [87].

Loss of integrin-dependent anchorage to the extracellular matrix leads to anoikis [88] or other mechanisms such as ATP deficiency due to reduced cellular glucose uptake [89]. The formation of microaggregates with platelets suppresses anoikis. Platelets activate cancer cell survival pathways (e.g. PI3K/AKT and MAPK signaling and RhoA-MYPT1-PP1-mediated YAP1 dephosphorylation) and, thereby, induce anoikis resistance and promote peritoneal and blood-borne metastasis [89,90]. The importance of YAP1 in this context has been confirmed in vivo, as a reduction of *YAP1* in cancer cells could protect against a thrombocytosis-induced increase in metastasis [90].

### 3.2. Evasion of the Immune System

Platelets can shield circulating tumor cells, and the capability of tumor cells to induce platelet aggregation correlates with their enhanced metastatic potential [91]. Accordingly, quantitative and/or qualitative defects in platelets reduce the number of metastasis [92,93].

A substantial amount of circulating tumor cells gets rapidly destroyed by NK cells. NK cells represent cytotoxic lymphocytes that play an important role in tumor immunosurveillance, preferentially eliminating targets with low or absent expression of major histocompatibility complex (MHC) class I and stress-induced expression of ligands for activating NK receptors. Studies involving the deletion of NK cells in mice provided evidence for the importance of this innate cellular subset for effective tumor rejection [94,95]. NK cells use a variety of activating and inhibitory receptors to recognize and eliminate malignant cells by secretion of cytolytic molecules such as granzyme B and perforin or IFN-γ or by exposure to the Fas ligand (FasL) and TNF-related apoptosis-inducing ligand (TRAIL) [96]. How the immune system senses tumor cells is still incompletely understood. Stress-induced ligands, such as those recognized by the activating immunoreceptor natural killer group 2, member D (NKG2D) on NK cells (and on subpopulations of T cells) or danger signals, directly released from transformed cells (e.g. HMGB1), and damaged tissue may represent mechanisms of tumor cell recognition [97,98].

Platelets and fibrin(ogen) interfere with the recognition of cancer cells by NK cells, thereby increasing the metastatic potential of tumor cells [99]. Platelet interaction with cancer cells leads to “pseudoexpression” of MHC class I molecules onto the surface of cancer cells via membrane fusion [100]. This renders metastatic cancer cells unrecognizable and leads to an impaired cytotoxicity as well as IFN-γ production by NK cells [100].

However, during tumor development, malignant cells follow several strategies to circumvent the antitumor activity of NK cells (Figure 3). Some tumors have the capacity to shed NKG2D ligands, such as MHC class I polypeptide-related sequence A (MICA) and MICB by metalloproteinase-mediated cleavage, resulting in reduced ligand on the surface of tumor cells [101]. Furthermore, tumor cells secrete immunomodulatory molecules that inhibit the activity of NK cells such as TGF-β, prostaglandin E_2_, adenosine or indoleamine 2,3-dioxigenase (IDO) [102], but it is yet unclear whether these molecules also play a role in tumor immune surveillance in the vasculature/circulatory system. Platelets also release TGF-β, which further weakens NK cell antitumor activity via downregulation of NKG2D on NK cells [103].

The effect(s) of other leukocytes in intravascular antitumor responses are less well-studied and understood. For example, neutrophils exert complex functions in this process, which depend on the phase of tumor development [83]. Early on, neutrophils accumulate in the premetastatic niche and inhibit metastasis by direct cytotoxic effects on malignant cells mediated by H_2_O_2_ production [104]. Platelet-neutrophil interactions enhance reactive oxygen production, thereby enhancing their cytotoxicity [2]. Later on, neutrophils, however, act in a prometastatic way, which includes the suppression of cytotoxic T [105] as well as NK cells [106].

The metastatic potential of a tumor depends also on the ability of tumor cells to swiftly extravasate into the surrounding tissue [83]. This step relies not only on interactions between tumor and endothelial cells but also requires interactions with leukocytes and platelets. This is not surprising, as platelets foster adhesion processes, and active adhesion, which is often tumor-type specific, seems to be essential for extravasation and metastasis.

Depletion of platelets or interference with their functions revealed a strong impact on tumor cell extravasation and further metastatic progression (summarized in [83]). A broad range of ligand-receptor pairs have been shown to impact on trans-endothelial migration/extravasation and, thus, affect metastasis. On the tumor-cell side, this includes extravasation-promoting molecules such as CCL2, angiopoietin like 4 (ANGPTL4) or podoplanin, whereas immune cells such as platelets produce TGF-β, CXCL5/7 and ATP and, thereby, target tumor cells, granulocytes/neutrophils and endothelial cells, respectively, which also induces extravasation of the tumor cells [83]. Formation of NETs, as mentioned earlier, not only impacts on tumor development but, also, promotes metastasis.

Metastasis is a complex multistage process, and metastatic dissemination of tumor cells, including the different processes mentioned above (detachment of the primary lesion, intravasation, escape of the immune system and persistence as circulating tumor cells and, finally, extravasation), is, to a high degree, cooperatively achieved with the epithelial-to-mesenchymal transition (EMT) [96,107]. EMT corresponds to a developmental regulatory program representing a cancer intrinsic capability by which transformed epithelial cells can acquire the abilities to invade, to resist apoptosis and to disseminate. By using a process involved in various steps of embryonic morphogenesis and wound healing, carcinoma cells can concomitantly acquire multiple attributes that enable invasion and metastasis [9]. Platelets promote EMT via direct cell-cell contact and aberrant cyclooxygenase (COX)-2 expression [108].

Although the inhibition of cancer cell intrinsic (or dependent) properties such as EMT and vascular alterations have dominated research for anticancer opportunities for a long time [109], recently, a change of paradigm has occurred that also takes the role of the immune system in cancer (and also metastasis) into account and highlights its strong impact. Inflammatory mediators such as cytokines and chemokines, as mentioned above, have direct effects on tumor cell- and immune cell functions, but they also contribute to the hallmarks of cancer, such as the facilitation of the EMT and increase of metastasis [18,110]. Apart from modulating immunosurveillance, platelets increase tumor growth and facilitate metastasis by inducing a mesenchymal-like phenotype in cancer cells [111] or by guiding the formation of an early metastatic niche [112], a process that involves TGF-β and direct interaction-needed NFĸB activation [111].

In the context of NK cells, EMT has been linked to the upregulation of multiple NKG2D ligands, including MICA, MICB and various UL16-binding proteins (ULBPs), along with the loss of NK cell inhibitory receptor (NKIR) ligands such as cadherin 1 (CDH1, also known as E-cadherin), turning potentially metastatic cells susceptible to recognition and elimination by NK cells [96].

Knockout studies on platelet molecules (such as NFE2, PAR4 and P-selectin) further refine the notion that platelets have crucial roles in tumor metastasis [113]. A reduced risk for cancer metastasis was also found in patients taking aspirin at doses sufficient to inhibit platelet function [114,115]. The role of aspirin in cancerogenesis seems to be of double action. Low-dose aspirin leads to long-lasting acetylation of COX-1, thereby preventing the synthesis of the prostanoids thromboxane A2 and prostaglandin E2 (PGE2), prostaglandin-containing oxidized phospholipids and sphingosine-1-phosphate [116]. These mediators contribute to the crosstalk among platelets, cancer cells and other cells of the tumor microenvironment. Several mechanisms have been postulated on how low-dose aspirin protects from early neoplastic transformation throughout the gastrointestinal tract: e.g. via downregulation of p-S6, a key regulator of the 40S ribosome biogenesis transcriptional program, in colorectal mucosa. This is closely related to tumor progression or via induction of several signaling pathways associated with a phenotypic switch of the stromal cells, which, in turn, results in the abnormal expression of COX-2 in epithelial cells and enhanced biosynthesis of the protumorigenic PGE2 [117]. Further, aspirin bears antimetastatic functions by averting the stem cell mimicry of cancer cells [118].

## 4. Conclusions and Outlook

For a long time, cancer research focused on the inhibition of cell-intrinsic or -dependent processes such as EMT and vascular alterations. Nowadays, more emphasis is increasingly put on the important role of the microenvironment. This also brings platelets and their interplay with leukocytes more into focus. More and more data emerge that platelets and their interactions with leukocytes are crucial at all stages of cancer. However, while platelets are well-explored in the context of hemostasis, thrombosis and vascular inflammation, the precise underlying mechanism and impact of platelet-leukocyte interactions in tumor development and progression still remains unknown. Thus, further research is warranted to bridge the gaps in connecting knowledge gained in other inflammatory diseases to cancer. Understanding platelets and their interplay with leukocytes and tumor cells could provide a basis for novel, targeted cancer therapies and will help us to understand the interplay of commonly used antiplatelet therapy and cancer development.

## Figures and Tables

**Figure 1 cells-09-00855-f001:**
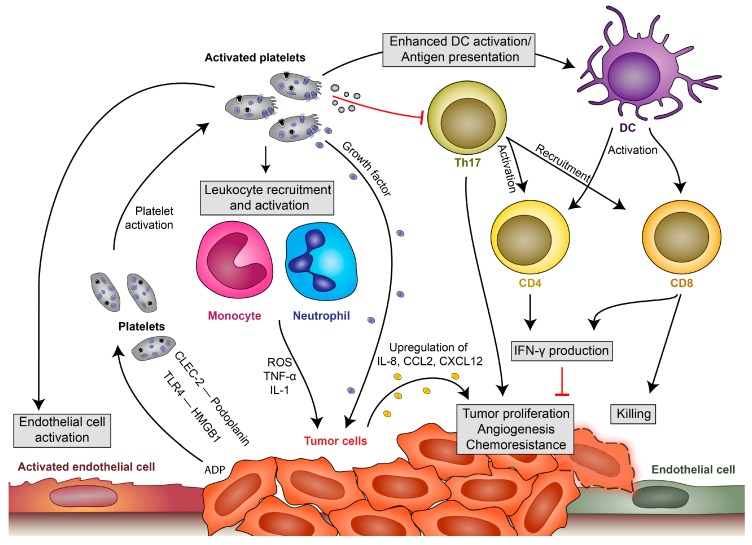
Platelet-leukocyte interactions during tumor development. Platelets become locally activated by the proinflammatory milieu or by tumor cells themselves (via C-type lectin-like immune receptor 2 (CLEC-2)–podoplanin interaction, TLR4 interaction with high-mobility group box 1 (HMGB1) or by release of ADP, which activates platelets via P2Y12). Activated platelets in turn release their growth factors and upregulate cytokine release by leukocytes, endothelial cells and, potentially, also tumor cells, which enhances local concentrations of interleukin (IL)-8, CCL2 and CXCL12 and leads to tumor proliferation, angiogenesis and chemoresistance. They enhance endothelial cell activation, which drives further cytokine release and recruitment of immune cells. Platelets also modulate immune responses by enhancing monocyte and neutrophil recruitment and activation. They have further been shown to enhance antigen presentation by dendritic cells (DCs), which fosters T-cell activation and results in enhanced interferon (IFN)-γ production and the killing of tumor cells by cytotoxic T cells. Platelets and their microvesicles have further been shown to inhibit activation of T-helper cell 17 (Th17) responses, which enhance tumor proliferation but can also enhance IFN-γ production and the killing of tumor cells.

**Figure 2 cells-09-00855-f002:**
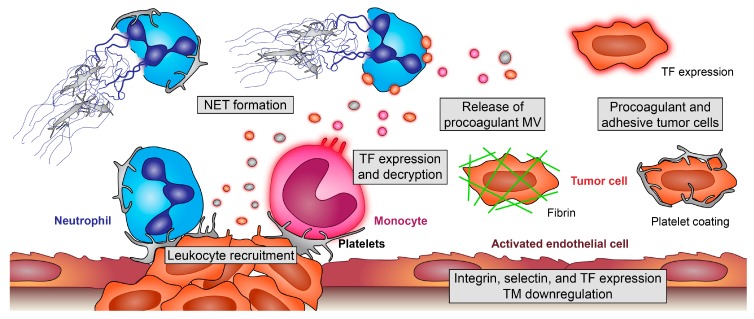
Pro-thrombotic effects of platelet-leukocyte interplay in cancer. Platelets enhance recruitment of monocytes and neutrophils and modulate their effector functions. Adherent monocytes become activated and release procoagulant, tissue factor (TF)-positive microvesicles (MV). Additionally, endothelial cells of the tumor microenvironment (TM) and tumor cells themselves can express TF and release TF-positive MV. Platelets boost these processes and foster decryption of inactive TF; platelet interaction with neutrophils enhances their recruitment and, also, the formation of procoagulant and prometastatic neutrophil extracellular traps (NETs), which ensnare and activate further platelets. Additionally, tumor cells can induce NET formation. On the surface of TF-positive tumor cells, coagulation factors become activated, and tumor cells encode themselves with fibrin, making them unrecognizable for immune cells and enhancing their stickiness at remote sites. Furthermore, platelets coat tumor cells, leading to their protection, enhanced metastasis but, also, increasing the risk for microthrombi formation and embolic events.

**Figure 3 cells-09-00855-f003:**
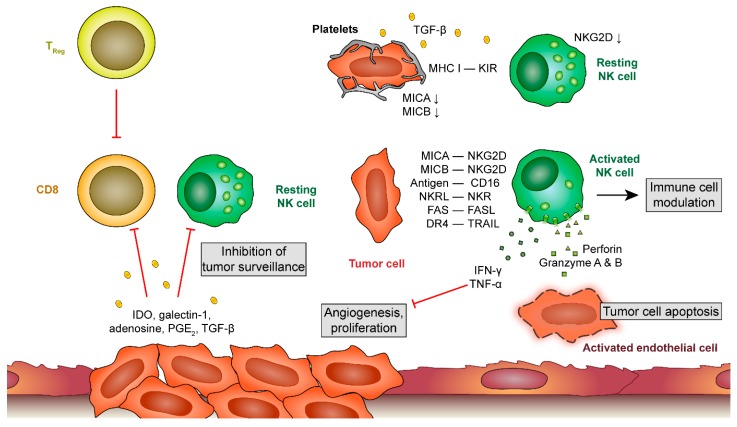
Platelet-leukocyte interactions during metastatic processes. At later stages of tumor development, tumor cells release a plethora of factors that inhibit cytotoxic responses of T cells and natural killer (NK) cells. When tumor cells detach and circulate in the blood stream, platelets provide an alternative mechanism to prevent recognition by NK cells. Unprotected tumor cells get recognized by NK cells, leading to their apoptosis by NK-derived perforin and granzyme A and B release. Platelets shield tumor cells by releasing tumor growth factor (TGF)-β which downregulates natural killer group 2, member D (NKG2D) on NK cells and by providing a major histocompatibility complex (MHC) class 1 receptor for killer cell immunoglobulin-like genes receptors (KIRs) on NK cells. Thereby, NK cells fail to become activated. IDO, indoleamine 2,3-dioxygenase; PGE 2, prostaglandin E2; MICA and MICB, MHC class I chain-related protein A and B; NKR(L), natural killer receptor (ligand); DR 4, death receptor 4; TRAIL, TNF-related apoptosis-inducing ligand.

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
