# Peer review of "Platelet-Leukocyte Interplay in Cancer Development and Progression"

_cells, 2020, doi:10.3390/cells9040855_

Round 1

Reviewer 1 Report

The review “Platelet-Leukocyte interplay in cancer development and progression” details a look at the current knowledge of the field and discusses several different aspects of platelet and leukocyte interactions in the context of tumor growth and metastasis. The authors touch on a wide variety of subjects and provide a clear understanding of what the processes occurring are in diseases. These range from tumorigenesis, to tumor growth, to thrombolytic events, and eventually tumor metastasis and highlight how platelets and leukocytes play a role in each step.

That being said, there are several issues that must be addressed in order for this manuscript to be more impactful. There is a substantial amount of work focused on platelet secretion affecting platelet-leukocyte interactions in cancer which was not addressed, see Kerr et al 2010.

  1. Broadly, many paragraphs are one-two sentences long. Readability could be vastly increased if paragraphs could be more consolidated and transitions made to be more congruent between topics.
  2. Overall, more mechanisms and pathways need to be mentioned or detailed more specifically.
  3. Section (1) could be more detailed by expanding on more specific interactions between platelets and leukocytes such as integrin activation as well general mechanosignaling mechanisms that modulate these interactions.
  4. Section (2.1) discusses some pattern recognition receptors. This should be expanded to include a few others that have been found to be similarly involved in tumor growth such as TLR2 or TLR3.
  5. Section (2.1) includes non-cellular aspects of platelet-leukocyte interactions but mainly the context of chemokines. This review would be more robust if there was a brief discussion of extracellular matrix components, both soluble and insoluble, and their interactions with platelets and leukocytes during tumor growth and formation.
  6. Section (2.2) lacks specific statements or cellular mechanisms in many instances. Inclusion of simple examples such as in section (2.1) would be beneficial.
  7. Section (2.3) contains some errors in nomenclature (e.g. the reference to TLR rather than which TLR the author is writing about) and inconsistent acronym defining (e.g. DIC (=disseminated intravascular coagulation) vs disseminated intravascular coagulation (DIC))
  8. A large portion of section (2.3) is dedicated to tissue factor. This section could be consolidated considerably and the inclusion of other factors in platelet-leukocyte interactions in thrombosis in cancer such as hyperactive integrin engagement via RGD-containing proteins.
  9. The inclusion of section (2.4) is mismatched in the context of the review theme as much of it is spent discussing immune system blockades. This section should be reworked to more heavily focus on the platelet-leukocyte interactions.
  10. Section (3.1) discusses anoikis and how platelets appear to mimic tumor cells in this capacity. Specific delineation of this mechanism should be stated.
  11. This review could be more impactful if a table of known metastatic cancers and platelet-leukocyte interactions could be more or at least if the diseases themselves were mentioned and catalogued more thoroughly.

Author Response

Reviewer 1

The review “Platelet-Leukocyte interplay in cancer development and progression” details a look at the current knowledge of the field and discusses several different aspects of platelet and leukocyte interactions in the context of tumor growth and metastasis. The authors touch on a wide variety of subjects and provide a clear understanding of what the processes occurring are in diseases. These range from tumorigenesis, to tumor growth, to thrombolytic events, and eventually tumor metastasis and highlight how platelets and leukocytes play a role in each step.

That being said, there are several issues that must be addressed in order for this manuscript to be more impactful. There is a substantial amount of work focused on platelet secretion affecting platelet-leukocyte interactions in cancer which was not addressed, see Kerr et al 2010.

  1. Broadly, many paragraphs are one-two sentences long. Readability could be vastly increased if paragraphs could be more consolidated and transitions made to be more congruent between topics.

We thank the reviewer for this comment and made changes throughout the manuscript.

  1. Overall, more mechanisms and pathways need to be mentioned or detailed more specifically.

We added more mechanistical insights throughout the manuscript, with emphasis on elaborating more on the topics mentioned in comments 3-11.

  1. Section (1) could be more detailed by expanding on more specific interactions between platelets and leukocytes such as integrin activation as well general mechanosignaling mechanisms that modulate these interactions.

We have added more information on this topic in Section 1.

  1. Section (2.1) discusses some pattern recognition receptors. This should be expanded to include a few others that have been found to be similarly involved in tumor growth such as TLR2 or TLR3.

As suggested, we have expanded the discussion on TLRs in Section 2.1. and mention some recent findings on TLR2 and TLR3.

  1. Section (2.1) includes non-cellular aspects of platelet-leukocyte interactions but mainly the context of chemokines. This review would be more robust if there was a brief discussion of extracellular matrix components, both soluble and insoluble, and their interactions with platelets and leukocytes during tumor growth and formation.

We now include extracellular matrix components in the discussion of Section 2.1.

  1. Section (2.2) lacks specific statements or cellular mechanisms in many instances. Inclusion of simple examples such as in section (2.1) would be beneficial.

We have now added several examples in Section 2.1.

  1. Section (2.3) contains some errors in nomenclature (e.g. the reference to TLR rather than which TLR the author is writing about) and inconsistent acronym defining (e.g. DIC (=disseminated intravascular coagulation) vs disseminated intravascular coagulation (DIC))

We thank the reviewer for pointing out these inconsistencies with nomenclature and abbreviations and have changed them.

  1. A large portion of section (2.3) is dedicated to tissue factor. This section could be consolidated considerably and the inclusion of other factors in platelet-leukocyte interactions in thrombosis in cancer such as hyperactive integrin engagement via RGD-containing proteins.

We have added integrins to Section 2.3.

  1. The inclusion of section (2.4) is mismatched in the context of the review theme as much of it is spent discussing immune system blockades. This section should be reworked to more heavily focus on the platelet-leukocyte interactions.

We have reworked Section 2.4. (which is now partly incorporated in Section 2.1) and now focus more on platelet-leukocyte interactions.

  1. Section (3.1) discusses anoikis and how platelets appear to mimic tumor cells in this capacity. Specific delineation of this mechanism should be stated.

We have now added details on the mechanism in Section 3.1.

  1. This review could be more impactful if a table of known metastatic cancers and platelet-leukocyte interactions could be more or at least if the diseases themselves were mentioned and catalogued more thoroughly.

We agree with the reviewer, however this is a very complicated task as some studies did not work directly with platelets but only on factors released by platelets and other studies were only performed in in vitro studies. To date there is not enough information available to catalogue the impact on different cancer types. We state this now more clearly in the review.

Reviewer 2 Report

In the present manuscript, Stoiber and Assinger reviewed the evidence on the involvement of platelet-leukocyte interplay in cancer.
The topic of this manuscript is clinically relevant. It is clearly written, and the figures are of excellent quality. However, I found some limitations that should be taken into account by the authors.

Major points

The authors completely ignored the contribution of prostanoids generated by platelets and by the interaction with leukocytes and cancer cells. This is a significant limitation because the clinical evidence of the role of platelets in cancer development is mainly coming from the data on the anticancer effects of aspirin, even at low doses, which mainly target the platelet COX-1 activity. This point has to be added to the review. In this context, they should add appropriate references, including Patrignani and Patrono JACC 2016 and Platelets 2018. Also, add two papers on the determinants of platelet-cancer cell interactions and metastasis development by Dovizio et al. Molecular Pharmacol 2013 and Guillem-Llobat et al. Oncotarget 2016.

Author Response

Reviewer 2

In the present manuscript, Stoiber and Assinger reviewed the evidence on the involvement of platelet-leukocyte interplay in cancer. 
The topic of this manuscript is clinically relevant. It is clearly written, and the figures are of excellent quality. However, I found some limitations that should be taken into account by the authors.

Major points

The authors completely ignored the contribution of prostanoids generated by platelets and by the interaction with leukocytes and cancer cells. This is a significant limitation because the clinical evidence of the role of platelets in cancer development is mainly coming from the data on the anticancer effects of aspirin, even at low doses, which mainly target the platelet COX-1 activity. This point has to be added to the review. In this context, they should add appropriate references, including Patrignani and Patrono JACC 2016 and Platelets 2018. Also, add two papers on the determinants of platelet-cancer cell interactions and metastasis development by Dovizio et al. Molecular Pharmacol 2013 and Guillem-Llobat et al. Oncotarget 2016.

We thank the reviewer for this important comment and have added these aspects (including references) to the manuscript. Please see changes marked in red in the chapter “3.1. Tumor dissemination” in paragraph 1 and 4.

Round 2

Reviewer 2 Report

The revised version is improved, and the Authors made most of the changes suggested by the Reviewers.